# Translating trait to state assessment: The case of grandiose narcissism

**Fien Heyde** [1]*, **Bart Wille**[1], **Evy Kuijpers**[2], **Jasmine Vergauwe**[1,2], **Joeri Hofmans**[2]

1 Department of Developmental, Personality, and Social Psychology, Ghent University, Gent, Belgium,
2 Department of Work and Organizational Psychology, Vrije Universiteit Brussel, Brussel, Belgium

* Fien.Heyde@UGent.be

## Abstract

Although interest in within-person variability in grandiose narcissism is growing, measurement tools are lacking that allow studying fluctuations in this personality characteristic in a differentiated manner (i.e., distinguishing narcissistic admiration and rivalry). This study explores whether a measurement approach using the six-item version of the Narcissistic Grandiosity Scale (NGS Rosenthal et al. (2007)) and six additional newly formulated adjectives allows assessing state admiration and rivalry. Structural characteristics and convergent validity of this approach were examined in an experience sampling study in which 114 adults participated, providing state assessments twice a day (total number of observations = 1306). Multilevel bifactor analyses revealed three factors (i.e., one general and two specific factors) at both within- and between-person levels. Further, admiration and rivalry showed a pattern of within-person associations with fluctuations in self-esteem and Big Five states that were consistent with theoretical expectations. Finally, average state admiration and average state rivalry correlated substantively with trait measures of these respective constructs assessed one week prior to the experience sampling design.

**Data Availability Statement:** The analysis code (Mplus syntaxes) for testing all models, data, and materials are available in the Open Science Framework (OSF) via this link: https://osf.io/srwbu/ DOI 10.17605/OSF.IO/SRWBU.

## Introduction

Grandiose narcissism is characterized by a grandiose self-concept, feelings of superiority, a strong need for power, a sense of entitlement, and a lack of empathy [2, 3]. It is a complex personality construct that is studied across various subdomains of psychology. Reviewing the grandiose narcissism literature, at least two important evolutions can be discerned. First, research is increasingly paying attention to the multifaceted nature of this construct [4]. Conceptualizing different "flavors" of narcissism may help to understand the contradictory effects of this trait on people's functioning in specific life domains, such as in romantic life [5] or at work [6]. Second, research is evolving from a static perspective in which narcissism–as a stable trait–is used to predict outcomes, to a more dynamic perspective in which narcissism is seen as fluctuating over the course of hours, days, and weeks [7]. This process-based approach is in line with a broader trend in personality research aimed at uncovering within-person dynamics in addition to the more traditional focus on between-person differences [8].

**Funding:** This study was funded by Fonds Wetenschappelijk Onderzoek (FWO), Belgium. https://www.fwo.be/ Funding received by BW and JV. Grant number: 3G000520 The funder played no role in the study design, data collection and analysis, decision to publish, or preparation of the manuscript.

**Competing interests:** The authors have declared that no competing interests exist.

Following these evolutions, the literature now witnesses particular challenges as regards the assessment of this construct. Specifically, the two trends described above call for assessment tools that are able to measure short-term fluctuations in narcissism *and* its subdimensions. As noted by Hortsmann and Ziegler [9], the construction of state level instruments is a delicate matter, and adopting established trait measures for this purpose is not always satisfactory. Although measures have been validated to assess both admiration (i.e., agentic/extraverted aspects of narcissism) and rivalry (i.e., antagonistic/disagreeable aspects of narcissism) at the trait or "between-person" level, namely the Narcissistic Admiration and Rivalry Questionnaire long-form (NARQ [3]) and short-form [10], we are unaware of a validated approach to measure these particular aspects at the level of fluctuating states (i.e., within-person level). In the current paper, we advocate that an adjective-based approach may fulfill the need for a comprehensive though differentiated state level assessment of grandiose narcissism because adjectives are proposed as the best way to assess personality states and allow to limit the response time within repeated measurement designs [11, 12]. In particular, a state-based assessment of both admirative and rivalrous facets of narcissism can be of interest, because nomological network analyses underscore the necessity to differentiate between these two facets. Admiration is interpreted as the adaptive form of narcissism while rivalry is rather seen as the maladaptive facet. Towards this end, we adopt one already validated instrument to represent the admiration aspect of narcissism (i.e., the Narcissistic Grandiosity Scale; NGS [13]), supplemented by a set of additional items specifically targeted at the rivalry component of grandiose narcissism. Before detailing this approach, an overview is provided of the multidimensional nature of this personality construct in order to frame the importance of a nuanced and multifaceted assessment.

## The multidimensional nature of narcissism

In a recent review, Sedikides [4] described narcissism as polyhedric, splitting the construct into various facets. At the level of the individual (versus the level of collective narcissism [14]), a distinction is first made between vulnerable and grandiose narcissism. Both domains contain a core of entitlement and antagonism [15]. Whereas vulnerable narcissism is associated with features such as psychological distress, hypersensitivity, low self-esteem, and neuroticism, grandiose narcissism, conversely, tends to go hand in hand with assertiveness, extraversion, and high self-esteem [16–19]. Further specification learns that the grandiose form of narcissism represents agentic and communal narcissism, two positively related but distinct subcomponents. This duality is manifested as self-enhancement strivings in the agentic (ambition, drive) versus communal (sociality, morality) domain. Finally, within the agentic domain of grandiose narcissism a distinction is made between admirative and rivalrous facets. Specifically, the Narcissistic Admiration and Rivalry Concept (NARC [3] [p1015]) describes how admiration and rivalry are two distinct but correlated facets of grandiose narcissism. They have the same overarching goal to maintain a grandiose self-concept while aiming to reach it via different social strategies. In admiration, aggrandizing the self-view is reached primarily through self-promotion, whereas in rivalry this self-view is maintained (or boosted) through diminishing others (i.e., self-protection). The strategy employed depends on the situation [20]. This mechanism can be best described as "if opportunity for promotion or demonstration of the grandiose, superior self, then self-affirm, self-promote, and self-enhance" for admiration and "if threat to own grandiosity and superiority, then strike back" for rivalry [21 p402].

## Fluctuations in narcissistic states

Research has recently begun to explore narcissism as a process, looking at within-person fluctuations in narcissistic states in addition to static between-person differences [7]. This

evolution aligns with a broader movement within personality psychology aimed at explaining meaningful intraindividual variation in human thoughts, feelings, and behavior across contexts and time [22]. This key area of personality research is built upon several prominent models that aim to: (a) conceptualize personality states (e.g., using parameters of personality *baseline*, *variability*, and *attractor* in the PersDyn model [23]); (b) clarify the link between states and traits (e.g., using the notion of density distributions [11]); and (c) explain the dynamic interaction between states and situational triggers (e.g., in terms of "if [Situation] then [Behavior]" contingencies [24]).

Whereas empirical research has now extensively documented the occurrence, predictors, and outcomes of fluctuations in Big Five personality constructs [25], studies looking at state level fluctuations in narcissism are less common to date. Using daily diary studies with undergraduate students, Giacomin and Jordan [7] were among the first to show that grandiose narcissism has a meaningful state component–that is, the observed within-person variability was not simply random error, as it related systematically to other psychological states and daily events. Next, Edershile and Wright [26] examined fluctuations in grandiose and vulnerable narcissism states and demonstrated how these processes related to dispositional assessments of both forms of narcissism. More recently, Mota and colleagues [27] examined within-person dynamics of agentic (i.e., admiration) and antagonistic (i.e., rivalry) aspects of state grandiose narcissism. Their findings demonstrated a substantial amount of both within- and between-person variation.

Together, these findings support a dynamic approach to narcissism because it varies substantially *within* subjects as well as *between* subjects, similar to many other personality constructs. There is hence a need for research on these narcissism fluctuations, as well as on its potential situational triggers. However, a closer look at this literature indicates some outstanding challenges, particularly concerning the measurement and modeling of these states.

### Operationalizing state level grandiose narcissism

Previous research on state level grandiose narcissism has generally adopted two possible assessment strategies. In a first approach, items are taken from the trait measure and the questionnaire is adapted by including "*right now*" in the instruction. Although this approach has proven successful when researching overall state level grandiose narcissism (e.g., using the 16-item version of the Narcissistic Personality Inventory; NPI [7, 28];), problems may arise for instruments designed to measure both admiration and rivalry. For NARQ, for instance, inspection of the items learns that for several of these (e.g., "*Mostly, I am very adept at dealing with other people*") simply adapting the instruction does not offer a viable solution. In a second approach, one can use adjectives to capture states on repeated occasions. To our knowledge, only one such adjective scale–i.e., the Narcissistic Grandiosity Scale (NGS [29])–has been suggested to use as a state measure and has been validated for the momentary assessment of (global) grandiose narcissism [13].

Until now, much less effort has been done to distinguish between admiration and rivalry when measuring state level grandiose narcissism. To our knowledge, there is only one study by Mota et al. [27] which used three items for state admiration and three items for state rivalry, whereby the entire itemset represented a blend of past behaviors (e.g., "Today, I was assertive"–*admiration*) and current feelings and thoughts (e.g., "I feel unacknowledged and criticized"–*rivalry*). Although their initial results were promising, the authors also indicate the need to explore a broader set of items covering especially the state level feelings associated with momentous expressions of both admiration and rivalry.

Given that the adjectives of the NGS have already been validated as a state level instrument for narcissistic grandiosity, this instrument can serve as a promising starting point for this matter. This approach does, however, require the NGS-itemset to be extended with additional adjectives, particularly adjectives reflecting the more rivalrous aspect of grandiosity.

## Modeling state level grandiose narcissism

In addition to the operationalization of grandiose narcissism and its two facets, one needs to consider how to best evaluate the empirical structure of this complex construct. In Back et al.'s [3] original work on trait admiration and rivalry, the approach consisted of modeling both facets as two correlated latent variables. However research on state level grandiose narcissism has typically not considered or tested the empirical structure of this construct. For instance, a study by Edershile et al. [13] confirmed that grandiose and vulnerable state level aspects can indeed be empirically separated, but no further distinction was investigated *within* grandiosity (see also Edershile and Wright [26]). Similarly, across different studies, Giacomin and Jordan [7, 30] modeled state grandiose narcissism as one factor. Finally, one exception is the work by Mota et al. [27] which acknowledged the facetted structure of state grandiosity by conducting separate analyses for admiration and rivalry. However, this study did not report on the structure of the measurement model underlying this differentiated measure of state grandiosity.

This brief review of the literature illustrates that little is currently known about the structure of state grandiosity, in particular the distinctiveness of rivalry and admiration, and how both aspects connect to the broader tendency of self-aggrandizement. Therefore, in addition to operationalizing both state components, a second central objective of the current study is to propose and test a modeling approach for state admiration and rivalry. For this purpose, we will depart from the theoretical NARC [3, 4, 20], which describes how both admiration and rivalry have the same overarching goal to maintain a grandiose self-concept while aiming to reach it via different social strategies.

## Current study

The overall objective of the current study is to propose and test a new measurement approach for state grandiosity and its two facets admiration and rivalry. As indicated above, this involves innovations at the level of the operationalization of this construct as well as with regard to the underlying measurement model.

**Starting from the NGS as a measure of grandiose admiration.**   In order to operationalize state grandiosity in a differentiated manner, the current study proposes using the adjectives included in the NGS as a starting point. Specifically, conceptual and empirical arguments can be made for using these NGS items to cover the aspect of narcissistic admiration in particular. Indeed, in the NARC the aspect of admiration has been described as a strive toward aggrandizing the self-view through self-promotion [3]. This connection between admiration and grandiosity is also evident when looking at the NGS-items (see Table 1), which all make reference to perceptions of admiration. Indeed, perceived admiration triggers the self-aggrandization strategy because when narcissists feel "glorious", for instance, they are invited to demonstrate their superior selves [20, 31].

Further, empirical support for this classification of the NGS-6 as a scale that captures admiration stems from previously reported associations between the NGS and the NARQ, i.e. the flagship inventory for narcissistic admiration and rivalry. Specifically, the NGS showed a correlation of $r = .73$ with narcissistic admiration, while the correlation with narcissistic rivalry

**Table 1. 12 adjectives for measuring state admiration and rivalry.**

|  | English | Dutch |
|---|---|---|
| Admiration (adopted from NGS) | Glorious | Groots |
|  | Envied | Benijd |
|  | Prestigious | Prestigieus |
|  | Brilliant | Briljant |
|  | Powerful | Invloedrijk |
|  | Superior | Superieur |
| Rivalry (new adjectives) | Contemptuous | Minachtend |
|  | Irritated | Geïrriteerd |
|  | Gleeful | Vervuld met leedvermaak |
|  | Envious | Jaloers |
|  | Resentful | Wraakzuchtig |
|  | Angry | Boos |

*Note.* NGS = Narcissism Grandiosity Scale

was $r = .31$ [3]. Finally, indirect empirical support comes from associations between NPI-scales on the one hand and NGS-6 and NARQ-scales on the other. As summarized in Table 2, the correlation pattern of the NGS with the NPI subdimensions [12] is highly similar to the correlation pattern between NARQ-admiration and NPI subdimensions, and a quite different pattern is observed for NARQ-rivalry [3] (see Table 2). For instance, whereas both the NGS-6 and the NARQ-admiration scale correlated .46 with the NPI Grandiosity/Exhibitionism dimension, the NARQ-rivalry scale correlated .18 with the same NPI dimension.

Because of these reasons, the current study proposes exploring the NGS-6 as a starting point for investigating state level variability in grandiose narcissism and its facets. Given the conceptual and empirical evidence reviewed above, the NGS-6 is specifically examined as a measure of grandiose admiration, whereas for rivalry an additional set of items is generated based on the NARC [3] (see Method section).

**The structure of state level grandiosity.** In order to obtain a better understanding of the structure of state grandiosity, the current study proposes examining this concept through the lens of a bifactor model [32]. Specifically, as illustrated in Fig 1, this approach allows specifying one general factor that reflects a stable component (i.e., overarching goal to maintain the grandiose self), next to two specific factors that represent occasion specific-influences (i.e., unique aspects of admiration and rivalry) [33]. In this way, the two strategies are able to predict variability beyond the general motive to maintain a grandiose self-concept. Although the bifactor model has been applied successfully to examine the structure of other multifaceted constructs (e.g., ADHD [34, 35]) this is, to the best of our knowledge, the first application of this model to grandiose narcissism.

**Table 2. Previously reported correlations of NGS-6 [12] and NARQ subdimensions [3] with NPI.**

|  | NPI | | |
|---|---|---|---|
|  | Leadership/Authority | Grandiosity/Exhibitionism | Entitlement/Exploitativeness |
| NGS-6 | .51 | .46 | .33 |
| NARQ Admiration | .47 | .46 | .26 |
| NARQ Rivalry | .19 | .18 | .47 |

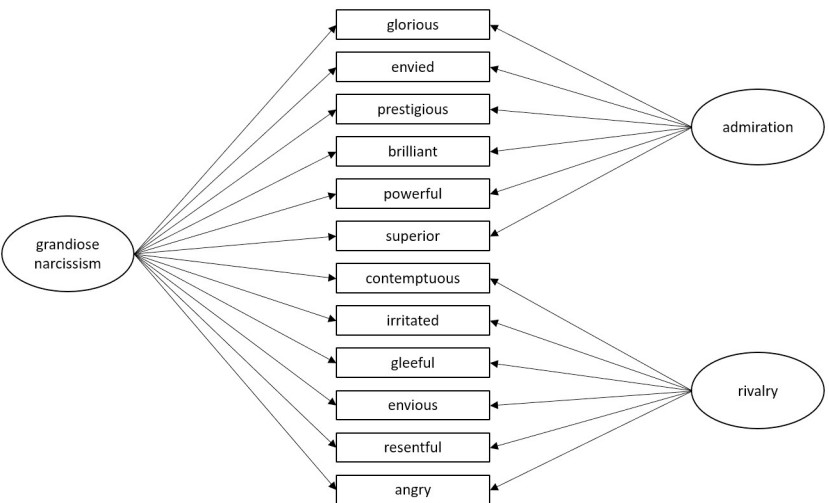

**Fig 1. The proposed bifactor model for narcissistic admiration and rivalry based on 12 adjectives for grandiose narcissism.**

## Research steps and hypotheses

The proposed approach for measuring state admiration and rivalry is investigated in several ways. First, as a starting point, we test the structure of the measurement model using multilevel bifactor modeling and compare it to previously used modeling strategies. Our general expectation here is that the model with one overarching grandiosity factor and two more specific factors (one for admiration and one for rivalry) will fit the data well, specifically at the within-person level which represents the state level.

Next, within-person associations between narcissism and other constructs from the nomological network are examined. Building on previous work on trait narcissism [3], we first focus on two personality domains relevant to narcissism research, namely self-esteem states and Big Five personality states. On the one hand, in line with this prior work at the trait level, we hypothesize that state narcissistic admiration will be positively associated with state self-esteem (H1) and state extraversion (H2). State narcissistic rivalry, on the other hand, is expected to be negatively associated with state self-esteem (H3) and state agreeableness (H4).

Finally, we also investigate the associations between trait and average state narcissism (i.e., between-person level). Drawing on the idea that the average state level is a meaningful descriptor of one's personality trait [11, 23], the following hypotheses concerning convergent validity are formulated: trait admiration is positively related to one's average state admiration (H5), and trait rivalry is positively related to one's average state rivalry (H6). In line with a recent study by Mota et al. [27], we expect correlation coefficients of .30 or higher.

## Materials and method

### Participants

The total sample consisted of 114 working adults (69.20% female; mean age = 36.81 years, *SD* = 12.79). Thirty-one percent of the participants had a master's degree, 25% had a professional bachelor's degree, and 24% had a secondary school degree. Other educational levels were less represented in this sample. The occupations held by the participants were highly

diverse. Participants completed 11.46 reports on average (57.30% response rate). The total number of observations is 1306.

## Procedure

Data were collected over a period of three weeks. Participants were recruited by two research associates. Potential participants received an invitation including a web link to the registration where they could leave their contact information. They were informed about the purpose of the study and were provided with the opportunity to raise questions. At the beginning of the study, participants completed an online baseline questionnaire assessing demographic characteristics and trait personality. Participants provided their informed consent prior to responding to the first questionnaire by going to the next page on the web page. One week later, the experience sampling study (ESM) started. For two weeks, participants received two text messages a day to fill out an online questionnaire programmed in *formr* [36]. These were sent at a random time during the workday (i.e., one in the morning and one in the afternoon). Participants were able to "snooze" these questionnaires for two hours. If they did not respond, they had to wait until the next text message. The studies involving human participants were reviewed and approved by the Ethical Commission of the Faculty of Psychological and Educational Sciences of Ghent University (application number: 2021/45).

## Measures

**Admiration and rivalry states.** As explained above, state admiration was operationalized through the NGS-6 [37], which three research associates translated independently from English into Dutch. Where needed, individual items were discussed among the translators in a second stage in order to agree upon a final version [38].

For state rivalry, six new adjectives were generated for the current study hereby closely relying on the work by Back et al. [3] who conceptualized rivalry as a striving for supremacy, devaluation of others, and aggressive behavior. Specifically, six items were formulated that characterize a situation wherein the grandiose self is threatened and the self-defense strategy would be activated [31]. We departed from the items incorporated in Back et al.'s [3] trait measure of rivalry and reformulated these into appropriate adjectives (see Table 1). Specifically, "contemptuous" resembles the NARQ item "*Other people are worth nothing*", "irritated" comes from "*I often get annoyed when I am criticized*", "gleeful" from "*I secretly take pleasure in the failure of my rivals*", "envious" from "*I can barely stand it if another person is at the center of events*", and "resentful" from "*I want my rivals to fail*". Finally, "angry" is an adjective proposed by the developers of the NARQ and reflects the aggressiveness within the rivalry construct.

**Big five personality states.** Big Five personality states were measured using a Dutch validated version of the 10-Item Personality Inventory (TIPI [39]). The TIPI uses adjectives as items and is therefore appropriate for ESM designs if the instruction for the participant is to indicate how they feel "right now". Each adjective was rated on a scale from 1 (*disagree strongly*) to 7 (*agree strongly*).

**Self-esteem states.** Self-esteem states were measured using a Dutch version of the Single Item Self-Esteem Scale (SISES [40]). Participants evaluated to which extent the statement "*I have high self-esteem*" applied to them "right now". This item was rated on a scale from 1 (*not very true of me*) to 5 (*very true of me*).

**Admiration and rivalry traits.** In the baseline questionnaire, participants completed a Dutch version of the NARQ [3] (retrieved from: http://www.persoc.net/Toolbox/Toolbox). The NARQ is an 18-item measure created to assess the two dimensions, admiration and

rivalry, at the trait level. Each item was rated on a six-point Likert scale from 1 (*disagree strongly*) to 6 (*agree strongly*).

## Results

The analysis code (Mplus syntaxes) for testing all models, data, and materials are available in the Open Science Framework (OSF) via this link.

### Structural analysis

We tested the structure of the narcissism state measure using confirmatory multilevel factor analysis in Mplus Version 8.4 [41]. Table 3 contains all Intraclass Correlation Coefficients (ICCs) of all items included in the model. The ICCs range from .54 to .84 indicating that 16 to 46% of the variation in the items is located at the within-person level. At the latent level, the ICC's are .79, .60, and .66 for the general factor, admiration and rivalry, respectively.

We modeled the item scores as being categorical, using the WLSMV estimator. Consistent with the conceptual idea that each item reflects variance due to a general grandiose narcissism factor but also due to one of two specific factors (i.e., admiration or rivalry), we ran a two-level bifactor model. Specifically, each of the twelve adjectives simultaneously loads on the general grandiose narcissism factor and on one of both specific factors. Moreover, the general factor is uncorrelated with the specific factors, while the specific factors are allowed to correlate. This bifactor model fitted the data relatively well ($\chi^2$ (83) = 111.43, CFI = .98, TLI = .97, RMSEA = .02; SRMR$_{within}$ = .05, SRMR$_{between}$ = .03). A nonsignificant negative residual variance was fixed to zero for model convergence issues. Because of this reason the degrees of freedom equal 83 rather than 82. Moreover, this model outperformed alternative models such as a model with a single factor at the within-person and a single factor at the between-person level ($\chi^2$ (108) = 1084.03, CFI = .42, TLI = .29, RMSEA = .08; SRMR$_{within}$ = .24, SRMR$_{between}$ = .11) and a model with two correlated factors at the within- and the between-person level ($\chi^2$ (106) = 431.32, CFI = .81, TLI = .76, RMSEA = .05; SRMR$_{within}$ = .14, SRMR$_{between}$ = .06).

Table 4 shows the standardized factor loadings for the two-level bifactor model for the within- and between-person level. All items show positive, statistically significant loadings on the general grandiose narcissism factor. Moreover, also for the admiration factor, all loadings are in the expected direction and statistically significant. For the rivalry factor, the adjectives "contemptuous" and "gleeful" had a nonsignificant factor loading at the within-person level, while this was the case for "contemptuous", "gleeful", "envious", and "resentful" at the between-person level. This suggests that those items do not share relevant variance with the other rivalry items beyond the variance they already share with all other grandiose narcissism items. Supplementary analyses using a partially saturated model with a saturated between-person (within-person) model is used to further investigate the goodness of within-person

**Table 3. Estimates of the intraclass correlations coefficients (ICC) of the items.**

| Item | ICC | Item | ICC |
|---|---|---|---|
| Glorious | .68 | Contemptuous | .75 |
| Envied | .72 | Irritated | .54 |
| Prestigious | .79 | Gleeful | .83 |
| Brilliant | .79 | Envious | .79 |
| Powerful | .76 | Resentful | .78 |
| Superior | .84 | Angry | .65 |

**Table 4. Results of the bifactor model.**

| Adjective | Factor loadings | | |
|---|---|---|---|
| | 1 (General) | 2 (Admiration) | 3 (Rivalry) |
| Within-person | | | |
| Glorious[a] | .33*** | .56*** | - |
| Envied[a] | .59*** | .23* | - |
| Prestigious[a] | .44*** | .51*** | - |
| Brilliant[a] | .38*** | .61*** | - |
| Powerful[a] | .30** | .61*** | - |
| Superior[a] | .32*** | .55*** | - |
| Contemptuous[b] | .64*** | - | .21 |
| Irritated[b] | .34** | - | .65*** |
| Gleeful[b] | .70*** | - | .05 |
| Envious[b] | .50*** | - | .32** |
| Resentful[b] | .41*** | - | .39*** |
| Angry[b] | .34* | - | .92*** |
| Between-person | | | |
| Glorious[a] | .56*** | .72*** | - |
| Envied[a] | .80*** | .30*** | - |
| Prestigious[a] | .52*** | .80*** | - |
| Brilliant[a] | .50*** | .85*** | - |
| Powerful[a] | .51*** | .65*** | - |
| Superior[a] | .65*** | .70*** | - |
| Contemptuous[b] | .95*** | - | -.07 |
| Irritated[b] | .79*** | - | .54*** |
| Gleeful[b] | .88*** | - | -.13 |
| Envious[b] | .76*** | - | .15 |
| Resentful[b] | 1.00*** | - | -.01 |
| Angry[b] | .75*** | - | .67*** |

*Note.*

* $p$ 05,

** $p < .01$,

*** $p < .001$.

[a]Admiration items,

[b]Rivalry items.

The Admiration and Rivalry factor correlate -.60 ($p < .001$) at the within-person level and -.04 ($p = .825$) at the between-person level.

(between-person) model fit [42]. Results indicate that the bifactor model fits the data well at both levels (see Appendix in S1 File).

Bifactor indices were calculated using Dueber's calculator [43] to further document the structural characteristics of this measurement approach. Table 5 shows all indices at both the within- and between-person level. Because of the focus on state level variation, we will only discuss the within-person level findings here.

First, omega (ω) is an index of internal reliability [44]. For the general factor, the total omega was .88, meaning that 88% of the variance in the total narcissism score can be attributed to the general factor. For the specific factors at the within-person level, omegaS is .82 for admiration and .84 for rivalry. Additionally, construct replicability (H) was calculated, which allows the evaluation of how well a latent variable is represented by a given set of items. The standard

**Table 5. Bifactor indices.**

|  | Omega/OmegaS | OmegaH/OmegaHS | H |
|---|---|---|---|
| Within-person |  |  |  |
| General factor | .88 | .56 | .79 |
| Admiration | .82 | .51 | .71 |
| Rivalry | .84 | .36 | .87 |
| Between-person |  |  |  |
| General factor | .98 | .80 | .99 |
| Admiration | .97 | .54 | .88 |
| Rivalry | .97 | .05 | .63 |

*Note.* OmegaS = omega subscale; omegaH = omega hierarchical; omegaHS = omega hierarchical subscale;
H = construct replicability.

criterium is that H values need to be .70 or above, meaning that at least 70% of the variability in the construct needs to be explainable by its own indicators [44, 45]. H is .79 for the general factor, .71 for admiration, and .87 for rivalry. The combination of ω and H indicates that the three factors are reliably measured and that the items represent the factors well.

Second, the bifactor indices also provide information about the multidimensionality of the construct. Omega hierarchical (omegaH or $\omega_H$) pertains to the proportion of variance in total scores that can be attributed to a single general factor while treating variability due to specific factors as measurement error [44]. OmegaH is .56, which indicates that total scores can be considered multidimensional (i.e., omegaH < .80) [46]. For the subscales (omegaHS or $\omega_{HS}$), these values are .51 for admiration and .36 for rivalry. This means that, after controlling for the general factor, the admiration subscale explains 51% of the variance, and the rivalry subscale explains 36%.

## Within-person associations between personality states

Table 6 shows the zero-order correlations between all personality states. Regarding within-person relationships, the findings confirmed our expectations by showing a positive association

**Table 6. Descriptive statistics and zero-order correlations for the study variables.** Within-person correlations are above, and between-person correlations are below the diagonal.

|  | M | SD between | SD within | 1 | 2 | 3 | 4 | 5 | 6 | 7 | 8 |
|---|---|---|---|---|---|---|---|---|---|---|---|---|
| 1. Rivalry | 1.57 | .72 | .41 | - | -.05 | -.22*** | -.29*** | -.11*** | .24*** | -.35*** | -.18*** |
| 2. Admiration | 2.37 | 1.04 | .47 | .49*** | - | .34*** | .34*** | -.01 | .17*** | .22*** | .28** |
| 3. Self-esteem | 3.11 | .85 | .54 | .03 | .44*** | - | .36*** | .01 | .26*** | .29*** | .24*** |
| 4. Extraversion | 4.84 | .91 | .81 | -.33*** | .21* | .37*** | - | .02 | .33*** | .34*** | .42*** |
| 5. Agreeableness | 4.42 | .67 | .52 | -.28** | -.31** | -.20* | .23* | - | .01 | .07** | .03 |
| 6. Conscientiousness | 5.65 | .95 | .54 | -.62*** | -.15 | .07 | .37*** | .19* | - | .25 | .30*** |
| 7. Emotional stab. | 5.25 | .90 | .73 | -.54*** | -.03 | .20* | .51*** | .09 | .53*** | - | .21*** |
| 8. Openness | 4.61 | .87 | .70 | -.29*** | .09 | .13 | .57*** | .01 | .28** | .48*** | - |

*Note.*

* $p < .05$,

** $p < .01$,

*** $p < .001$

**Table 7. Zero-order correlations between average state and trait narcissism (NARQ).**

|  | 1 | 2 | 3 | 4 |
|---|---|---|---|---|
| 1. Average state admiration | - |  |  |  |
| 2. Average state rivalry | .49*** | - |  |  |
| 3. Trait admiration | **.53***** | .19* | - |  |
| 4. Trait rivalry | .47*** | **.59***** | .45*** | - |

*Note.*

* *p* < .05,

** *p* < .01,

*** *p* < .001

between state admiration and state self-esteem (H1), and a negative association between state rivalry and state self-esteem (H3). Also, the associations with the Big Five personality states confirm our hypotheses. State admiration is positively correlated to extraversion (H2), and state rivalry is negatively related to agreeableness (H4). Regarding the remaining Big Five personality states, state admiration had a positive correlation with conscientiousness, emotional stability, and openness. There was no association between state admiration and agreeableness. State rivalry had a negative correlation with extraversion, emotional stability, and openness, and a positive correlation with conscientiousness.

### Associations between trait and average state narcissism

Finally, we examined the relationships between average state narcissism and trait narcissism measured through the initial survey (see Table 7). The coefficients indicating convergent validity were higher than the expected level of *r* = .30 (see coefficients in bold). In addition, the results showed that average state admiration is also highly correlated with trait rivalry, while average state rivalry has only a modest association with trait admiration.

Next, we computed partial correlations to estimate the degree to which the relation between state admiration (rivalry) and trait admiration (rivalry) is independent of their relation with trait rivalry (admiration). Partial correlations provided further support for the convergent validity of our measurement approach. The partial correlation between average state admiration and trait admiration, while controlling for trait rivalry was .40 (*p* < .001), while the partial correlation between average state rivalry and trait rivalry, while controlling for trait admiration equaled .58 (*p* < .001). This is substantially higher than the partial correlation between average state admiration and trait rivalry while controlling for trait admiration (*r* = .29, *p* < .001) and the partial correlation between average state rivalry and trait admiration, while controlling for trait rivalry (*r* = -.13, *p* = .198).

Finally, to delve further into the nature of the six state rivalry items, we also investigated the correlation between trait rivalry and average state rivalry, while controlling for trait emotional stability. The partial correlation between both rivalry constructs was .55 (*p* < .001), indicating that the relationship between trait and average state rivalry cannot be explained by a general tendency to experience negative affect.

### Discussion

Research on narcissism has begun to move beyond investigating the structural components toward exploring narcissism as a process. However, there is currently a lack of measurement instruments to capture state narcissism in a multifaceted way. The current research started to

address this gap by exploring the momentary assessment of narcissistic admiration and rivalry using a validated measurement instrument for state narcissistic grandiosity, the NGS-6 [13, 37], as a starting point. After scrutinizing the NGS-6 items against a literature review on grandiose narcissism, the NGS-6 was used as a base for measuring state narcissistic admiration, and six new adjectives were used for assessing state narcissistic rivalry.

The ICCs, based on both observed and latent variables, demonstrate the presence of state variability within individuals. The between-person variability is greater than the within-person variability, which aligns with earlier studies indicating that the within-person variability is substantial [7, 47] but not as pronounced as in other personality domains such as extraversion [11]. To test the structural characteristics of this measurement approach consisting of 12 adjectives, we departed from theory (e.g., NARC [3]) and assumed a bifactor model capturing one general (grandiosity) factor and two specific factors (for admiration and rivalry). Our results at the within-person level show that this model performed better than a one- and two-factor model, indicating that these state items are able to capture variation in general grandiose narcissism as well as unique variation in admiration and rivalry. However, as is common when fitting bifactor models, a number deviations from the expected model could also be observed [33]. Specifically, one residual variance was fixed to zero and inspection of (within-person) factor loadings also indicated that two rivalry items, i.e., "contemptuous" and "gleeful", did not share relevant variance with the other rivalry items beyond the variance they already share with all other grandiose narcissism items. To further investigate the psychometric properties of this measurement approach, we examined the internal reliability and magnitude of the three identified factors. The obtained bifactor indices provide evidence for the reliability of these factors, meaning that they are well-represented by the items. Additionally, the indices indicate that the general factor is indeed multidimensional and that the specific factors are sizeable. Based on the theoretical justification for using a bifactor model, the good fit relative to alternative models, and the results of the bifactor indices, we conclude that the bifactor model is appropriate for modeling admiration and rivalry at the within-person level.

Next, we proceeded by inspecting the within-person associations between grandiose narcissism and related personality states. Departing from the nomological network of admiration and rivalry at the trait level [3, 10], we expected a similar association at the state level, in which the NGS-6 would align with the nomological network of trait admiration, whereas the new state rivalry adjectives would align with the nomological network of trait rivalry. First, in line with the trait literature [3] and confirming the idea of "puffed-up but shaky selves" [48], state admiration showed a positive relation to self-esteem, while state rivalry was negatively related to self-esteem. As the antagonistic nature of rivalry may lead to social conflict that comes along with ego threats [3, 31], it is reasonable that these rivalry states are more likely to be accompanied by low self-esteem at the momentary level. Second, consistent with our expectations, we found a positive association between admiration and extraversion, as well as a negative association between rivalry and agreeableness at the within-person level. Moreover, the associations between state admiration and the remaining Big Five personality states are further in line with Back et al. [3]. Regarding state rivalry, our findings also largely resemble Back et al.'s [3] associations with the Big Five. In addition to the negative association with agreeableness, we found a negative correlation between rivalry and extraversion that was similar to Back et al. [3] (i.e., $r = -.11/-.24$). Next, state rivalry shows a substantially stronger relation with emotional stability, which is unexpected based on Back et al. [3] ($r = -.19$ between rivalry and neuroticism). The six new items were formulated based on the definition of narcissistic rivalry–including a striving for supremacy, devaluation of others, and aggressive behavior–, which is also reflected in the NARQ-rivalry itemset [3]. It is remarkable, however, that this itemset resembles the Narcissistic Vulnerability Scale (NVS [13, 49]) to some extent (e.g., *envious* and *resentful*).

Consequently, one could argue that the new items partially tap into neurotic or vulnerable narcissism, next to rivalrous narcissism. Although the conceptual discussion of how vulnerable narcissism relates to the rivalrous aspect of grandiose narcissism falls beyond the scope of the present paper, it is relevant to note at this point that previous research has also pointed out this connection ($r$ = .57) [3]. In addition, the new adjectives describe emotions that contain an *approach tendency* instead of an avoidance tendency [50]. Relevant in this regard is that vulnerable narcissism is associated with social withdrawal [51], which is in line with the idea that vulnerable narcissism comes into play when the individual failed to restore the narcissistic esteem and when there is no perceived chance for further retaliation [31]. On the contrary, narcissistic rivalry comes into play after experiencing ego-threat, after which narcissistic individuals then try to restore the grandiose self by defending themselves. In other words, the approach tendency incorporated within the adjectives is probably more suitable for narcissistic rivalry than for vulnerable narcissism. However, future research focusing simultaneously on–even more–different flavors of state narcissism (e.g., vulnerable, grandiose, communal, agentic, rivalry, admiration) [4] is needed to test this assumption and clarify this conceptual discussion. Lastly, inconsistent with Back et al. [3], the current study yielded a positive association between state rivalry and conscientiousness. It needs to be further examined whether the specific context in which participants were assessed, i.e., work, can account for such a pattern.

Finally, associations were examined between trait narcissism and the average state admiration and rivalry. Although the average state score for both facets was indeed related to the corresponding trait score, our findings revealed a high association between the average state admiration score and trait rivalry, while the average state rivalry was only modestly related to trait admiration. Moreover, even after controlling for trait admiration, the correlation between average state admiration and trait rivalry remained substantial. According to Wetzel et al. [52], narcissistic individuals can be divided into two groups: those characterized by admiration and those characterized by admiration *and* rivalry. This suggests that people high on trait rivalry score higher on both state rivalry and admiration, while people high on trait admiration will score higher on state admiration, but do not necessarily score higher on state rivalry. As the last step, partial correlations between trait rivalry and average state rivalry also indicated that this association was not driven by a general tendency to experience negative emotionality.

## Limitations and future research

Although the measurement approach presented in this study was generally supported by the data at hand, a number of inconsistencies were also identified that can be addressed by future studies. Specifically, whereas it was clear from the different analyses that the existing NGS-6 items can indeed be successfully used to tap into state admiration, the measurement of state rivalry requires some further attention. Although the overall bifactor model fitted the data relatively well and despite sufficiently high factor analytic reliability estimates for this subscale, the pattern of bifactor loadings also indicated that additional items need to be formulated that better tap into this particular narcissism aspect (after accounting for the variance shared with the general narcissism factor). In light of the conceptual discussion outlined above, this will require paying close attention to potential overlap with related constructs, such as vulnerable narcissism.

## Conclusion

This study aimed to emphasize the importance of a validated measurement instrument for assessing state narcissism rather than using trait measures. Therefore, we proposed and tested a new measurement approach for state grandiose narcissism. Consistent with theoretical

perspectives, we showed that a bifactor model, which comprises narcissism factors that encompass both shared and unique narcissism aspects, can indeed be specified to capture within-person fluctuations in this multifaceted personality characteristic. Although the overall model fitted the data relatively well, a number of specific measurement challenges were also identified that require further attention, particularly regarding the assessment of state rivalry. To conclude, we trust that this article provides evidence that investigating alternative measurement models, in particular a bifactor model, can be advantageous in the realm of state narcissistic admiration and rivalry.

## Supporting information

**S1 File. Appendix: Partially saturated models.**
(PDF)

## Author Contributions

**Conceptualization:** Fien Heyde, Bart Wille, Jasmine Vergauwe.

**Data curation:** Fien Heyde.

**Formal analysis:** Fien Heyde, Evy Kuijpers, Joeri Hofmans.

**Funding acquisition:** Bart Wille, Jasmine Vergauwe.

**Investigation:** Fien Heyde.

**Methodology:** Fien Heyde.

**Project administration:** Fien Heyde.

**Resources:** Fien Heyde.

**Supervision:** Bart Wille, Jasmine Vergauwe, Joeri Hofmans.

**Validation:** Fien Heyde.

**Visualization:** Fien Heyde.

**Writing – original draft:** Fien Heyde, Bart Wille.

**Writing – review & editing:** Fien Heyde, Bart Wille, Jasmine Vergauwe, Joeri Hofmans.

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
