## [Decision Letter · Decision Letter 0]

7 Feb 2023

PONE-D-22-35155Translating trait to state assessment: The case of grandiose narcissismPLOS ONE

Dear Dr. Heyde,

Thank you for submitting your manuscript to PLOS ONE. After careful consideration, we feel that it has merit but does not fully meet PLOS ONE’s publication criteria as it currently stands. Therefore, we invite you to submit a revised version of the manuscript that addresses the points raised during the review process. I find the suggestion to include ICC very relevant, esp. in the light of the most recent New Scholars podcasthttps://www.youtube.com/watch?v=LbIBa7R5Yeg

While you can exercise licencia poética, I would suggest to address narcissism as a trait and things one level lower as facets.

We look forward to receiving your revised manuscript.

Kind regards,

Frantisek Sudzina

Academic Editor

PLOS ONE

Journal Requirements:

Reviewers' comments:

Reviewer's Responses to Questions

**Comments to the Author**

1. Is the manuscript technically sound, and do the data support the conclusions?

Reviewer #1: Yes

Reviewer #2: Partly

2. Has the statistical analysis been performed appropriately and rigorously? 

Reviewer #1: Yes

Reviewer #2: Yes

3. Have the authors made all data underlying the findings in their manuscript fully available?

Reviewer #1: Yes

Reviewer #2: Yes

4. Is the manuscript presented in an intelligible fashion and written in standard English?

Reviewer #1: Yes

Reviewer #2: Yes

5. Review Comments to the Author

Reviewer #1: This study examined the dynamic association between narcissistic admiration and rivalry (along with Big Five traits and self-esteem) in a community sample in Belgium via experience sampling, data from which they analyzed using bifactor multi-level modeling. Results suggest that narcissistic admiration and rivalry adhere to a bifactor structure, and that there are generally strong associations between state and trait narcissism. Findings further show differential associations between state admiration and rivalry and other constructs measures (Big Five traits and self-esteem). Strengths of this study include solid study design and advanced data analytic approach. Weaknesses are minor and largely include some inefficiencies in writing.

1. This paper is generally very well written, although there is some room for improvement. For example, the authors note “the current study” multiple times in the Introduction (e.g., p. 6, 8), which leads to reader to think that they are about to describe the current study (but they do not).

2. The authors also discuss the use of bifactor modeling in the Introduction, noting how it has not yet been applied to narcissism. This is not an optimal way of framing the study for several reasons. First, it would be more useful to discuss what a bifactor model does at the level of constructs rather than statistical (i.e., what it would mean for narcissistic grandiosity to have the characteristics amenable to analysis using bifactor modeling). Second, just because something has not been done before is not a good reason to do it.

3. The Results section is rather long, spanning almost 8 pages (pp. 12-20). The authors should be more concise and cut this by at least half. For example, the discussion of omega (pp. 15-16) could be briefer. The authors also need not repeat coefficients already listed in tables (e.g., the correlations on p. 19).

4. The authors seem to state that a limitation of their study is that it uses a community sample (vs. a college sample) and that “more research is needed to evaluate the generalizability of our findings across samples and contexts” (p. 24). Although it is certainly true that more research would be useful, this seems an unnecessary statement unless the authors had specific samples in mind in which they would like to study this construct further. If not, they could delete this sentence, as it is not very meaningful.

Reviewer #2: In their article “Translating trait to state assessment: The case of grandiose narcissism”, the authors have developed a measurement instrument to capture grandiose narcissism at the state level. The instrument aims to capture short-term fluctuation in both facets of grandiose narcissism, namely admiration and rivalry. The authors present data from an experience sampling study in support of the construct validity of the developed scale.

Researchers are increasingly advocating that personality, such as narcissism, should be examined at the state level, next to the traditional trait level. Validated state scales remain scarce despite the enthusiasm for process-based approaches in personality research. The scarcity of state measurements emphasizes the scientific relevance of the current study. Next to the innovative approach and relevance of the presented study, I recommend that several issues need to be addressed before publication. I have sorted my comments into major and minor issues.

Major issues:

I appreciate the conciseness of the article but suggest adding the following sections to better embed the article’s findings in previous research. First, a brief overview of models that have been used to model grandiose narcissism in the past (ideally, at the state and trait level) . Following this, the models should then be systematically tested against one another. It should be clear, why the authors chose the current model of grandiose narcissism. Second, a more in-depth review of past studies that have examined grandiose narcissism. For example, the study from Mota et al. (2022) appears to be particularly relevant because it also assessed admiration and rivalry states. So far, this study is only briefly cited for the expected strength of the associations between trait and aggregated state scores (i.e., H5 & H6), leaving it an open question why Mota et al.’s study has not informed the item formulation or in which ways the current study expands Mota et al.’s findings.

Given that the article entails results at two level of analysis, I recommend using precise language to distinguish the within-person (WP) and between-person (BP) level from each other. For example, on p. 9, the authors state that they expect a bifactor model to fit the data well but without explicitly stating that the same structural model is assumed at both levels of analysis (which is not self-evident given that associations of variables can differ across levels of analysis, e.g. Molenaar, 2004). Moreover, the authors use the term ‘state’ most of the time to refer to the WP level, but sometimes also to refer to the BP level. For example, on p. 19 the heading for the section in which results between trait narcissism and state narcissism at the BP level (i.e., using aggregated state scores) reads “association between trait and state narcissism”. To enhance clarity, it could be helpful to provide a brief statement that state scores can be used to model associations at the WP and BP level, and then introduce the terminology that will be used throughout the paper.

I suggest considering adding the following analyses: First, estimates of the ICC (ideally at the latent level) to gain insights into how much of the variation in the items can be attributable to within-person fluctuations (i.e., the major focus of the manuscript). Second, I recommend using partially saturated models (Ryu & West, 2009) to evaluate the models separately at each level of analysis since otherwise misfit at the BP level could be masked by the results of the WP level (due to the large difference in sample size at both levels).

I do not fully agree with the following conclusions drawn from the data and ask the authors to provide additional arguments or to reconsider their conclusions. First, the article concludes that the bifactor model fits the data well. Yet, although negative variances and non-significant factor loadings of the specific factors are not uncommon for bifactor models (Eid et al., 2017), such results question the validity of the model. Second, on p. 15, the following interpretation is provided “For the general factor at the within-person level, the total omega was .88, meaning that 88% of the variance in the total narcissism score can be attributed to the general factor, and 12% to random error”. Yet, omega does not provide estimates about the variance that goes back to random error but how much of the variance of the items can be accounted for by the general factor. In other words, it is unknown to what the 12% are attributable. Third, on p. 20 it is concluded that “[…] the relation between state admiration and trait rivalry is fully explained by trait admiration” although the partial correlation between state admiration and trait rivalry after accounting for trait admiration is r =.29. Fourth, on p. 21/22 it is stated that “[…] in line with our expectations, admiration was positively related to extraversion, while rivalry was accompanied by disagreeableness at the within-person level”. Yet, extraversion correlated in comparable magnitude with admiration (r = .34) and rivalry (r = -.29), suggesting that this association is not as specific as expected.

Lastly, I feel that the conclusion section could benefit from more concrete recommendations for future research while acknowledging that this may be my subjective perception.

Minor issues:

P. 12: I would add the term ‘confirmatory’ to multilevel factor analysis

P. 23: “Finally, associations were examined between trait narcissism and the average within-person fluctuations in admiration and rivalry”: To my knowledge, the term ‘within-person fluctuations’ is either used to refer to single states or to the individual standard deviation but not to refer to the individual mean which is meant in this section. Thus, I would rather speak about average or aggregated state scores.

I would also suggest speaking of different narcissism facets or domains, instead of “forms”

References

Eid, M., Geiser, C., Koch, T., & Heene, M. (2017). Anomalous results in G-factor models: Explanations and alternatives. Psychological methods, 22(3), 541–562. https://doi.org/10.1037/met0000083

Molenaar, P. C. M. (2004). A Manifesto on Psychology as Idiographic Science: Bringing the Person Back Into Scientific Psychology, This Time Forever. Measurement: Interdisciplinary Research and Perspectives, 2(4), 201–218. https://doi.org/10.1207/s15366359mea0204_1

Mota, S., Mielke, I., Kroencke, L., Geukes, K., Nestler, S., & Back, M. (2022). Daily dynamics of grandiose narcissism: distribution, stability, and trait relations of admiration and rivalry states and state contingencies. European Journal of Personality, 0(0). https://doi.org/10.1177/08902070221081322

Ryu, E., & West, S. G. (2009). Level-Specific Evaluation of Model Fit in Multilevel Structural Equation Modeling. Structural Equation Modeling: A Multidisciplinary Journal, 16(4), 583–601. https://doi.org/10.1080/10705510903203466

6. PLOS authors have the option to publish the peer review history of their article (what does this mean?). If published, this will include your full peer review and any attached files.

Reviewer #1: No

Reviewer #2: **Yes: **Anabel Büchner

---

## [Author Response · Author response to Decision Letter 0]

3 Mar 2023

Response to reviewers

Thank you for submitting your manuscript to PLOS ONE. After careful consideration, we feel that it has merit but does not fully meet PLOS ONE’s publication criteria as it currently stands. Therefore, we invite you to submit a revised version of the manuscript that addresses the points raised during the review process.

I find the suggestion to include ICC very relevant, esp. in the light of the most recent New Scholars podcast

https://www.youtube.com/watch?v=LbIBa7R5Yeg

We followed this suggestion and added the ICCs to our manuscript.

While you can exercise licencia poética, I would suggest to address narcissism as a trait and things one level lower as facets.

To enhance comprehension, we referred to admiration and rivalry as facets. When we refer to the between-person level we speak of traits or average states and for the within-person level, we speak of states.

Journal Requirements:

Reviewers' comments:

1. Is the manuscript technically sound, and do the data support the conclusions?

Reviewer #1: Yes

Reviewer #2: Partly

2. Has the statistical analysis been performed appropriately and rigorously? 

Reviewer #1: Yes

Reviewer #2: Yes

3. Have the authors made all data underlying the findings in their manuscript fully available?

Reviewer #1: Yes

Reviewer #2: Yes

4. Is the manuscript presented in an intelligible fashion and written in standard English?

Reviewer #1: Yes

Reviewer #2: Yes

5. Review Comments to the Author

Reviewer #1: This study examined the dynamic association between narcissistic admiration and rivalry (along with Big Five traits and self-esteem) in a community sample in Belgium via experience sampling, data from which they analyzed using bifactor multi-level modeling. Results suggest that narcissistic admiration and rivalry adhere to a bifactor structure, and that there are generally strong associations between state and trait narcissism. Findings further show differential associations between state admiration and rivalry and other constructs measures (Big Five traits and self-esteem). Strengths of this study include solid study design and advanced data analytic approach. Weaknesses are minor and largely include some inefficiencies in writing.

1. This paper is generally very well written, although there is some room for improvement. For example, the authors note “the current study” multiple times in the Introduction (e.g., p. 6, 8), which leads to reader to think that they are about to describe the current study (but they do not).

Thank you for this comment. We reread our manuscript, but we are not sure we understand this comment. When we speak of “the current study”, we do refer to our study and describe the central aim or what we will conduct in our study. 

On page 7 we describe what we aim to investigate. Therefore, when we talk about the current study, we do describe our study.

“As there is currently no available adjective-based measure to assess narcissistic admiration and rivalry, and considering that the adjectives of the NGS primarily relate to affect, the central objective of the present study is to ascertain the extent to which this short grandiosity scale can be used to gauge more specific aspects of state level grandiose narcissism, in particular state admiration and state rivalry.”

On page 9, we state that we propose to explore the NGS-6 as a starting point because of the arguments above. Consequently we explain that we will complete the questionnaire with new rivalry items. 

“Against this background, the current study proposes exploring the NGS-6 as a starting point for investigating state-level variability in grandiose narcissism and its subfacets. Given the conceptual and empirical evidence reviewed above, the NGS-6 is specifically examined as a measure of grandiose admiration, whereas for rivalry an additional set of items will be generated (see Method section).”

So unless we misinterpreted your comment, we conclude that when we speak of “the current/present study”, we do describe our study.

2. The authors also discuss the use of bifactor modeling in the Introduction, noting how it has not yet been applied to narcissism. This is not an optimal way of framing the study for several reasons. First, it would be more useful to discuss what a bifactor model does at the level of constructs rather than statistical (i.e., what it would mean for narcissistic grandiosity to have the characteristics amenable to analysis using bifactor modeling). Second, just because something has not been done before is not a good reason to do it.

We appreciate this comment as it gives us the opportunity to provide a more in-depth argumentation for the use of the bifactor model. We agree that solely “because something is never done before” is not a good argument to conduct a new study. However, we used bifactor modelling because it offers an innovative opportunity to enhance our theoretical understanding of the concept of grandiose narcissism. Narcissistic admiration and rivalry are two distinct but correlated facets of grandiose narcissism. They have the same goal to maintain a grandiose self, while reaching it through other strategies. Admiration uses self-promotion and rivalry maintains a grandiose self by self-defence (Back et al., 2013; Grapsas et al., 2020). The situation determines which strategy can be used to maintain the grandiose self. This mechanism can be best described as “if opportunity for promotion or demonstration of the grandiose, superior self, then self-affirm, self-promote, and self-enhance” for admiration and “if threat to own grandiosity and superiority, then strike back” for rivalry (Morf et al., 2011, p. 402).

Using a bifactor model, we are able to model the multidimensionality of grandiose narcissism. Within a bifactor model, the general factor represents a stable component (i.e., the goal to maintain the self), while the specific factors represent occasion-specific influences (i.e., admiration and rivalry; Eid et al., 2017). In this way, the two strategies are able to predict variability beyond the general motive to maintain a grandiose self-concept.

We added some extra explanation on the choice of the bifactor model in our manuscript on page 9-10.

“In the current study, based on the theoretical Narcissistic Admiration and Rivalry Concept (NARC; Back et al., 2013; Grapsas et al., 2020; Sedikides, 2021), which describes how both admiration and rivalry have the same overarching goal to maintain a grandiose self-concept while aiming to reach it via different social strategies, we propose examining this concept through the lens of a bifactor model (Reise et al., 2010). Specifically, as illustrated in Figure 1, this approach allows specifying one general factor that reflects a stable component (i.e., overarching goal to maintain the grandiose self), next to two specific factors that represent occasion specific-influences (i.e., unique aspects of admiration and rivalry; Eid et al., 2017). In this way, the two strategies are able to predict variability beyond the general motive to maintain a grandiose self-concept. Although the bifactor model has been applied successfully to examine the structure of other multifaceted constructs (e.g., ADHD; Gomez et al., 2018; Martel et al., 2011) this is, to the best of our knowledge, the first application of this model to grandiose narcissism.”

3. The Results section is rather long, spanning almost 8 pages (pp. 12-20). The authors should be more concise and cut this by at least half. For example, the discussion of omega (pp. 15-16) could be briefer. The authors also need not repeat coefficients already listed in tables (e.g., the correlations on p. 19).

Thank you for bringing up this comment. We agree that the results section is rather long. However, for people not familiar with these more advanced/novel techniques, such as the bifactor indices, it seems important to explain the meaning of all the parameters. Nevertheless, we reconsidered this section and shortened the part on bifactor indices and partial correlations. We also removed the correlation coefficients of the association between admiration/rivalry and the Big Five that were not incorporated in the hypotheses.

Further we would like to enlighten why the results are important for providing an answer to our hypotheses. First, the structural analyses help to answer our first question on the structure of modeling state admiration and rivalry. In order to verify the bifactor model we need to compare its fit to the fit of previously used models and have to discuss the factor loadings. The bifactor indices provide more important information on internal reliability and multidimensionality. Second, for hypotheses 1-4 we discuss the within-person correlation coefficients between personality states. For the benefit of the reader, we would like to repeat the correlation coefficients in the text. However, we agree that the correlation coefficients of the association with the other personality states not included in the hypotheses do not need to be repeated. Finally, hypotheses 5 and 6 are answered by the results described under associations between trait and average state narcissism. Although the partial correlations could be considered as extra analyses, we believe that they have a crucial meaning in testing the hypotheses.

4. The authors seem to state that a limitation of their study is that it uses a community sample (vs. a college sample) and that “more research is needed to evaluate the generalizability of our findings across samples and contexts” (p. 24). Although it is certainly true that more research would be useful, this seems an unnecessary statement unless the authors had specific samples in mind in which they would like to study this construct further. If not, they could delete this sentence, as it is not very meaningful.

Thank you for this suggestion. We used a community sample of employees because of the relevance of narcissism within the work context. Our statement was very broadly formulated and indeed therefore not very meaningful. Consequently, we followed your suggestion and removed this limitation from our manuscript.

Reviewer #2: In their article “Translating trait to state assessment: The case of grandiose narcissism”, the authors have developed a measurement instrument to capture grandiose narcissism at the state level. The instrument aims to capture short-term fluctuation in both facets of grandiose narcissism, namely admiration and rivalry. The authors present data from an experience sampling study in support of the construct validity of the developed scale.

Researchers are increasingly advocating that personality, such as narcissism, should be examined at the state level, next to the traditional trait level. Validated state scales remain scarce despite the enthusiasm for process-based approaches in personality research. The scarcity of state measurements emphasizes the scientific relevance of the current study. Next to the innovative approach and relevance of the presented study, I recommend that several issues need to be addressed before publication. I have sorted my comments into major and minor issues.

Major issues:

I appreciate the conciseness of the article but suggest adding the following sections to better embed the article’s findings in previous research. First, a brief overview of models that have been used to model grandiose narcissism in the past (ideally, at the state and trait level) . Following this, the models should then be systematically tested against one another. It should be clear, why the authors chose the current model of grandiose narcissism. Second, a more in-depth review of past studies that have examined grandiose narcissism. For example, the study from Mota et al. (2022) appears to be particularly relevant because it also assessed admiration and rivalry states. So far, this study is only briefly cited for the expected strength of the associations between trait and aggregated state scores (i.e., H5 & H6), leaving it an open question why Mota et al.’s study has not informed the item formulation or in which ways the current study expands Mota et al.’s findings.

Thank you for your suggestion. In the previous version of our manuscript we already compared the bifactor model with a one-factor and a two-factor model. We chose for the bifactor model because of the theoretical basis and the good fit in comparison with other models. In the revised manuscript we clarified which measurement models were tested in previous research on page 9.

“In addition to the operationalization of state grandiose narcissism and its subfacets, one needs to consider how to best evaluate the empirical structure of this complex construct at the state level (i.e., within-person level). Originally, Back et al. (2013) modeled trait admiration and rivalry as two correlated latent variables. At the state level, Edershile et al. (2019) first tested the structure of grandiose and vulnerable narcissism as a multilevel two-factor model, and consequently modeled grandiose narcissism as a model with one factor at both the within- and between-person level (Edershile & Wright, 2020). Similarly, Giacomin and Jordan (2014, 2016a) modeled state grandiose narcissism as one factor. Finally, Mota et al. (2022) did not report on the structure of the measurement model, but conducted separate analyses for state admiration and rivalry.”

Previous research on state narcissism has mainly used trait measurements with an adapted instruction. An alternative approach is using an adjective-based instrument such as the Narcissistic Grandiosity Scale. Mota et al. (2022) developed new items on the cognitive, affective, and behavioural aspects of narcissistic admiration and rivalry. They indicate that although this is a good approach, there is a need for assessing narcissistic feelings in more detail. As there is currently no adjective-based measure available to assess narcissistic admiration and rivalry, and considering that the adjectives of the NGS primarily relate to affect, our study meets the need formulated by Mota et al. (2022). We added more information on previous studies measuring state narcissism on page 6.

“Previous research on state-level grandiose narcissism has generally adopted two possible assessment strategies. In a first approach, items are taken from the trait measure and the questionnaire is adapted by including “right now” in the instruction. Although this approach has proven successful in research measuring overall state-level grandiose narcissism (e.g., using the a 16-item version of the Narcissistic Personality Inventory; NPI; Raskin & Terry, 1988; see Giacomin & Jordan, 2016), problems may arise for instruments designed to measure both admiration and rivalry. For NARQ, for instance, inspection of the items learns that for several of these (e.g., “Mostly, I am very adept at dealing with other people”) simply adapting the instruction does not offer a viable solution. In a second approach, one can use adjectives to capture states on repeated occasions. To our knowledge, only one such adjective scale – i.e., the Narcissistic Grandiosity Scale (NGS; Rosenthal et al., 2020) – has been suggested to use as a state measure and has been validated for the momentary assessment of grandiose narcissism (Edershile et al., 2019). 

Regarding narcissistic admiration and rivalry, there is only one study by Mota et al. (2022) that intended to measure both facets at the state level within an undergraduates sample. State admiration and rivalry were both assessed using the aggregated score of three items covering the cognitive, affective and behavioral aspects. For admiration the items were: “I feel special and admired”, “cheerful”, and “Today, I was assertive”. The rivalry items were: “I feel unacknowledged and criticized”, “annoyed”, and “Today, I was hostile”. Although the results were in line with their expectations, the researchers indicate the importance of including a broader set of state measures that cover narcissistic feelings.

As there is currently no adjective-based measure available to assess narcissistic admiration and rivalry, and considering that the adjectives of the NGS primarily relate to affect, the central objective of the present study is to explore in what manner and to what extent this short grandiosity scale can be used to gauge more specific aspects of (state-level) grandiose narcissism, in particular (state) admiration and (state) rivalry. At the conceptual level, grandiosity can be directly linked to admiration, which is about aggrandizing the self-view through self-promotion (Back et al., 2013). This is also evident when looking at the NGS-items (see Table 1), which all make reference to perceptions of admiration. Indeed, as noted by Back (2018), perceived admiration triggers the self-aggrandization strategy because when narcissists feel “glorious”, for instance, they are invited to demonstrate their superior selves.”

Finally, we also mentioned the study of Mota et al. (2022) when discussing previous research on state narcissism on page 5-6.

“Whereas empirical research has now extensively documented the occurrence, predictors, and outcomes of fluctuations in Big Five personality constructs (e.g., Debusscher et al., 2017), studies looking at state-level fluctuations in narcissism are less common to date. Using daily diary studies with undergraduate students, Giacomin and Jordan (2016) were among the first to show that grandiose narcissism has a meaningful state component – that is, the observed within-person variability was not simply random error, as it related systematically to other psychological states and daily events. Next, Edershile and Wright (2020) examined fluctuations in grandiose and vulnerable narcissism states and demonstrated how these processes related to dispositional assessments of both forms of narcissism. More recently, Mota and colleagues (2022) examined within-person dynamics of agentic (admiration) and antagonistic (rivalry) aspects of state grandiose narcissism. their findings demonstrated a substantial amount of both within- and between-person variation.”

Given that the article entails results at two level of analysis, I recommend using precise language to distinguish the within-person (WP) and between-person (BP) level from each other. For example, on p. 9, the authors state that they expect a bifactor model to fit the data well but without explicitly stating that the same structural model is assumed at both levels of analysis (which is not self-evident given that associations of variables can differ across levels of analysis, e.g. Molenaar, 2004). Moreover, the authors use the term ‘state’ most of the time to refer to the WP level, but sometimes also to refer to the BP level. For example, on p. 19 the heading for the section in which results between trait narcissism and state narcissism at the BP level (i.e., using aggregated state scores) reads “association between trait and state narcissism”. To enhance clarity, it could be helpful to provide a brief statement that state scores can be used to model associations at the WP and BP level, and then introduce the terminology that will be used throughout the paper.

Thank you for this comment and mentioning the irregularities in our terminology. We indeed did not specify on which level we expected the bifactor model. Given the focus of this manuscript on state narcissism, we focus on the within-person level in the description of our results. For instance, the bifactor indices discussed in the manuscript are all about the within-person level. Therefore, we expect that the bifactor model fit the data well, specifically at the within-person level. 

Based on your comment and suggestion, the manuscript is now adjusted accordingly. When we refer to the within-person level, we speak of “states” and for the between-person level we use “average state”. Further, we specified that we expect the bifactor model to fit well at the within-person level.

I suggest considering adding the following analyses: First, estimates of the ICC (ideally at the latent level) to gain insights into how much of the variation in the items can be attributable to within-person fluctuations (i.e., the major focus of the manuscript). Second, I recommend using partially saturated models (Ryu & West, 2009) to evaluate the models separately at each level of analysis since otherwise misfit at the BP level could be masked by the results of the WP level (due to the large difference in sample size at both levels).

Thank you for this suggestion. We agree that it is beneficial to include ICCs in our results. We added the ICC’s of the observed items as well as the ICCs on the latent level. The ICCs of the items indicate that 16 to 46% of the variation is located at the within-person level, while the ICCs on the latent level show that 21-40% of the variation is located at the within-person level.

These findings align with previous studies by Giacomin & Jordan (2016a, 2016b) indicating that the within-person variability in narcissism is substantial but not as high as in other personality domains such as extraversion (Fleeson, 2001).

The ICCs are added to the manuscript and we compare them to previous research in the discussion on page 21.

“The ICCs, based on both observed and latent variables, demonstrate the presence of state variability within individuals. Although the between-person variability is greater than the within-person variability, this aligns with earlier studies indicating that the within-person variability is substantial (Giacomin & Jordan, 2016a, 2016b) but not as pronounced as in other personality domains such as extraversion (Fleeson, 2001).”

Table 3

Estimates of the intraclass correlations coefficients (ICC) of the items 

Item ICC Item ICC

Glorious .68 Contemptuous .75

Envied .72 Irritated .54

Prestigious .79 Gleeful .83

Brilliant .79 Envious .79

Powerful .76 Resentful .78

Superior .84 Angry .65

We did not think of the potential effects of misfit at the between-level before. Therefore, we want to thank you for the suggestion to use partially saturated models, as it gave us the opportunity to further investigate our model. First, we want to emphasize that the focus of the study lays on the within-person level. This implies that potential misfit at the between-level would not endanger our conclusions regarding measuring state grandiose narcissism. However, given the fact that misfit at the between-person level could influence the reliability of the estimates at the within-person level, we agree that we needed to further investigate the model.

As a first step, we looked at the fit measures of the bifactor model which was already reported in our original manuscript. Besides general fit measures such as CFI, TLI and RMSEA, Mplus also generates the Standardized Root Mean Square Residual (SRMR) separately for the within- and between-person level. These indicate that at both levels, the bifactor model fits the data relatively well. Nevertheless, in a second step, we also ran partially saturated models. The resulting factor loadings of both partially saturated models are completely comparable with our bifactor model. Specifically, the partially saturated model including a fully saturated model at the between-person level resulted in the same factor loadings at the within-person level and the one with a fully saturated model at the within-person level resulted in the same estimated factor loadings at the between-person level.

We added a footnote on page 15 in our manuscript and the results of the partially saturated models are included in the appendix. Mplus syntaxes and output can be found on the Open Science Framework.

“2 Supplementary analyses using a partially saturated model with a saturated between-person (within-person) model is used to further investigate the goodness of within-person (between-person) model fit (Ryu & West, 2009). Results indicate that the bifactor model fits the data well at both levels (see Appendix).”

I do not fully agree with the following conclusions drawn from the data and ask the authors to provide additional arguments or to reconsider their conclusions. 

(1) First, the article concludes that the bifactor model fits the data well. Yet, although negative variances and non-significant factor loadings of the specific factors are not uncommon for bifactor models (Eid et al., 2017), such results question the validity of the model. 

(2) Second, on p. 15, the following interpretation is provided “For the general factor at the within-person level, the total omega was .88, meaning that 88% of the variance in the total narcissism score can be attributed to the general factor, and 12% to random error”. Yet, omega does not provide estimates about the variance that goes back to random error but how much of the variance of the items can be accounted for by the general factor. In other words, it is unknown to what the 12% are attributable. 

(3) Third, on p. 20 it is concluded that “[…] the relation between state admiration and trait rivalry is fully explained by trait admiration” although the partial correlation between state admiration and trait rivalry after accounting for trait admiration is r =.29. 

(4) Fourth, on p. 21/22 it is stated that “[…] in line with our expectations, admiration was positively related to extraversion, while rivalry was accompanied by disagreeableness at the within-person level”. Yet, extraversion correlated in comparable magnitude with admiration (r = .34) and rivalry (r = -.29), suggesting that this association is not as specific as expected.

Thank you for the critical evaluation of our conclusions. We agree that we made some misinterpretations and we now added more nuance to our manuscript.

(1) Regarding the bifactor model, there are indeed some deviations such as a negative residual variance and some non-significant loadings for the specific factors. Although these problems are common when fitting a bifactor model (Eid et al., 2017), this indicates that there are still some challenges.

Nevertheless, despite these issues, our results indicate that the bifactor model has a relatively good fit when compared to other models such as a one- and two-factor model. Additionally, the bifactor indices provide evidence for the reliability of the factors and multidimensionality of the general factor. Together with the theoretical justification of the bifactor model, we conclude that, despite the deviations, the bifactor model is a relatively good model for our data. Nevertheless, we added more nuance to our discussion on page 21-22.

“To test the structural characteristics of this measurement approach consisting of 12 adjectives, we departed from theory (e.g., NARC; Back et al., 2013) and assumed a bifactor model capturing one general (narcissism) factor and two specific factors (for admiration and rivalry). Our results at the within-person level show that this model performed better than a one- and two-factor model, indicating that these state items are able to capture variation in general grandiose narcissism as well as unique variation in admiration and rivalry. However, as is common when fitting bifactor models, a number of deviations from the expected model could also be observed (Eid et al., 2017). Specifically, one residual variance was fixed to zero and inspection of (within-person) factor loadings also indicated that two rivalry items, i.e., “contemptuous” and “gleeful”, did not share relevant variance with the other rivalry items beyond the variance they already share with all other grandiose narcissism items. To further investigate the psychometric properties of this measurement approach, we examined the internal reliability and magnitude of the three identified factors. The obtained bifactor indices provide evidence for the reliability of these factors, meaning that they are well-represented by the items. Additionally, the indices indicate that the general factor is indeed multidimensional and that the specific factors are sizeable. Based on the theoretical justification for using a bifactor model, the good fit relative to alternative models, and the results of the bifactor indices, we conclude that the bifactor model is appropriate for modeling admiration and rivalry at the within-person level.”

(2) Thank you for noticing our misinterpretation of omega. We changed this in the manuscript.

(3) We appreciate the feedback on the partial correlation between state admiration and trait rivalry. Since the correlation is still .29 (p < .001) our conclusion is indeed incorrect. The association between state admiration and trait rivalry is only partially explained by trait admiration. This finding, however, is consistent with the results of Wetzel et al. (2016), which were already addressed in our discussion section. There appear to be two kinds of narcissists: those characterized by admiration, and those characterized by both admiration and rivalry. This suggests that people high on trait rivalry score high on both state admiration and rivalry. This translates into the association between state admiration and trait rivalry, not being entirely explained by trait admiration.

We incorporated this finding in our discussion (p. 24). 

“Finally, associations were examined between trait narcissism and the average state admiration and rivalry. Although the average state score for both subtypes was indeed related to the corresponding trait score, our findings revealed a high association between the average state admiration score and trait rivalry, while the average state rivalry was only modestly related to trait admiration. Moreover, even after controlling for trait admiration, the correlation between average state admiration and trait rivalry remained substantial. According to Wetzel et al. (2016), narcissistic individuals can be divided into two groups: those characterized by admiration and those characterized by admiration and rivalry. This suggests that people high on trait rivalry score higher on both state rivalry and admiration, while people high on trait admiration will score higher on state admiration, but do not necessarily score higher on state rivalry.”

(4) We only speak of the relation between admiration and extraversion because we recapitulate our hypotheses. Unfortunately, we overlooked the relationship between rivalry and extraversion in the discussion of the other relationships with the Big Five. The negative association is consistent with the findings of Back and colleagues (2013; i.e., r = -.11/-.24). We added this now to our discussion (p. 23).

“Second, consistent with our expectations, we found a positive association between admiration and extraversion, as well as a negative association between rivalry and agreeableness at the within-person level. Moreover, the associations between state admiration and the remaining Big Five personality states are further in line with Back et al. (2013). Regarding state rivalry, our findings largely resemble Back et al.’s (2013) associations with the Big Five. In addition to the negative association with agreeableness, we found a negative correlation between rivalry and extraversion that was similar to Back et al. (2013) (i.e., r = -.11/-.24).”

Lastly, I feel that the conclusion section could benefit from more concrete recommendations for future research while acknowledging that this may be my subjective perception.

We appreciate this comment and would like to highlight two concrete recommendations. First, this study aimed to emphasize the importance of developing validated measurement instruments for the momentary assessment of grandiose narcissism. This questions the approach adopted in several previous studies which consisted of simply taking the items from a trait measure and solely adapting the instruction by including “right now”. This approach is problematic for several items such as “Mostly, I am very adept at dealing with other people” (NARQ; Back et al., 2013). Therefore, we explored the use of an adjective-based instrument, the NGS, for creating a validated state measurement instrument for narcissistic admiration and rivalry. Second, we wanted to highlight the possibility of using an alternative modeling technique that can be theoretically linked to the Narcissistic admiration and rivalry concept, namely the bifactor model. Although there are still some challenges, this model outperformed previously used models. In conclusion, we recommend that future research further investigates the use of a bifactor model in the context of state narcissistic admiration and rivalry.

We added these recommendations to our conclusion on page 23:

“This study aimed to emphasize the importance of a validated measurement instrument rather than using trait measures. Therefore, we proposed and tested a new measurement approach for state grandiose narcissism. Consistent with theoretical perspectives, we showed that a bifactor model, which comprises narcissism factors that encompass both shared and unique narcissism aspects, can indeed be specified to capture within-person fluctuations in this multifaceted personality characteristic. Although the overall model fitted the data relatively well, a number of specific measurement challenges were also identified that require further attention, particularly regarding the assessment of state rivalry. To conclude, we trust that this article provides evidence that investigating alternative measurement models, in particular a bifactor model, can be advantageous in the realm of state narcissistic admiration and rivalry.”

Minor issues:

P. 12: I would add the term ‘confirmatory’ to multilevel factor analysis

This was adjusted in our manuscript.

P. 23: “Finally, associations were examined between trait narcissism and the average within-person fluctuations in admiration and rivalry”: To my knowledge, the term ‘within-person fluctuations’ is either used to refer to single states or to the individual standard deviation but not to refer to the individual mean which is meant in this section. Thus, I would rather speak about average or aggregated state scores.

Thank you for noticing. We adjusted this in our manuscript.

I would also suggest speaking of different narcissism facets or domains, instead of “forms”

We adjusted this throughout our manuscript.

References

Eid, M., Geiser, C., Koch, T., & Heene, M. (2017). Anomalous results in G-factor models: Explanations and alternatives. Psychological methods, 22(3), 541–562. https://doi.org/10.1037/met0000083

Molenaar, P. C. M. (2004). A Manifesto on Psychology as Idiographic Science: Bringing the Person Back Into Scientific Psychology, This Time Forever. Measurement: Interdisciplinary Research and Perspectives, 2(4), 201–218. https://doi.org/10.1207/s15366359mea0204_1

Mota, S., Mielke, I., Kroencke, L., Geukes, K., Nestler, S., & Back, M. (2022). Daily dynamics of grandiose narcissism: distribution, stability, and trait relations of admiration and rivalry states and state contingencies. European Journal of Personality, 0(0). https://doi.org/10.1177/08902070221081322

Ryu, E., & West, S. G. (2009). Level-Specific Evaluation of Model Fit in Multilevel Structural Equation Modeling. Structural Equation Modeling: A Multidisciplinary Journal, 16(4), 583–601. https://doi.org/10.1080/10705510903203466

References

Back, M. D., Küfner, A. C. P., Dufner, M., Gerlach, T. M., Rauthmann, J. F., & Denissen, J. J. A. (2013). Narcissistic admiration and rivalry: Disentangling the bright and dark sides of narcissism. Journal of Personality and Social Psychology, 105(6), 1013–1037. https://doi.org/10.1037/a0034431

Eid, M., Geiser, C., Koch, T., & Heene, M. (2017). Anomalous results in G -factor models: Explanations and alternatives. Psychological Methods, 22(3), 541–562. https://doi.org/10.1037/MET0000083

Fleeson, W. (2001). Toward a structure- and process-integrated view of personality: Traits as density distributions of states. Journal of Personality and Social Psychology, 80(6), 1011–1027. https://doi.org/10.1037/0022-3514.80.6.1011

Giacomin, M., & Jordan, C. H. (2016a). The wax and wane of narcissism: Grandiose narcissism as a process or state. Journal of Personality, 84(2), 154–164. https://doi.org/10.1111/jopy.12148

Giacomin, M., & Jordan, C. H. (2016b). Self-focused and feeling fine: Assessing state narcissism and its relation to well-being. Journal of Research in Personality, 63, 12–21. https://doi.org/10.1016/j.jrp.2016.04.009

Grapsas, S., Brummelman, E., Back, M. D., & Denissen, J. J. A. (2020). The “Why” and “How” of Narcissism: A Process Model of Narcissistic Status Pursuit. Perspectives on Psychological Science, 15(1), 150–172. https://doi.org/10.1177/1745691619873350

Morf, C. C., Horvath, S., & Torchetti, L. (2011). Narcissistic self-enhancement: Tales of (successful?) self-portrayal. In M. D. Alicke & C. Sedikides (Eds.), Handbook of self-enhancement and self-protection (pp. 399–424). The Guilford Press. https://psycnet.apa.org/record/2011-04015-019

Mota, S., Mielke, I., Kroencke, L., Geukes, K., Nestler, S., & Back, M. (2022). Daily dynamics of grandiose narcissism: Distribution, stability, and trait relations of admiration and rivalry states and state contingencies. European Journal of Personality. https://doi.org/https://doi.org/10.1177/08902070221081322

Wetzel, E., Leckelt, M., Gerlach, T. M., & Back, M. D. (2016). Distinguishing subgroups of narcissists with latent class analysis. European Journal of Personality, 30(4), 374–389. https://doi.org/10.1002/PER.2062

6. PLOS authors have the option to publish the peer review history of their article (what does this mean?). If published, this will include your full peer review and any attached files.

Do you want your identity to be public for this peer review? For information about this choice, including consent withdrawal, please see our Privacy Policy.

Reviewer #1: No

Reviewer #2: Yes: Anabel Büchner

---

## [Decision Letter · Decision Letter 1]

20 Mar 2023

PONE-D-22-35155R1Translating trait to state assessment: The case of grandiose narcissismPLOS ONE

Dear Dr. Heyde,

Thank you for submitting your manuscript to PLOS ONE. After careful consideration, we feel that it has merit but does not fully meet PLOS ONE’s publication criteria as it currently stands. Therefore, we invite you to submit a revised version of the manuscript that addresses the points raised during the review process.

The manuscript requires a minor revision. All the remining comments raised by both reviewers (see below) need to be addressed fully and clearly in order for the manuscript to be accepted.

We look forward to receiving your revised manuscript.

Kind regards,

Srebrenka Letina, Ph.D.

Academic Editor

PLOS ONE

Journal Requirements:

Additional Editor Comments:

The authors need to address clearly and fully all remaining comments from the reviewers.

Reviewers' comments:

Reviewer's Responses to Questions

**Comments to the Author**

1. If the authors have adequately addressed your comments raised in a previous round of review and you feel that this manuscript is now acceptable for publication, you may indicate that here to bypass the “Comments to the Author” section, enter your conflict of interest statement in the “Confidential to Editor” section, and submit your "Accept" recommendation.

Reviewer #1: (No Response)

Reviewer #2: All comments have been addressed

2. Is the manuscript technically sound, and do the data support the conclusions?

Reviewer #1: Yes

Reviewer #2: Yes

3. Has the statistical analysis been performed appropriately and rigorously? 

Reviewer #1: Yes

Reviewer #2: Yes

4. Have the authors made all data underlying the findings in their manuscript fully available?

Reviewer #1: Yes

Reviewer #2: Yes

5. Is the manuscript presented in an intelligible fashion and written in standard English?

Reviewer #1: Yes

Reviewer #2: Yes

6. Review Comments to the Author

Reviewer #1: This revised study took a bifactor modeling approach to examining the dynamic association between narcissistic admiration and rivalry, finding that narcissistic admiration and rivalry adhere to a bifactor structure, that there are generally strong associations between state and trait narcissism, and that there are differential associations between state admiration and rivalry and other constructs (e.g., self esteem). The authors were somewhat responsive to my and the other Reviewer’s comments, resulting in a somewhat improved manuscript. Two of my original comments that the authors did not adequately address remain.

1. My first comment is a clarification of my first comment on the original submission (which was not clear to the authors). Rather than mentioning “the current/present study” (and it is clear that the authors are referring to this study and not some other study) multiple times throughout the Introduction, just mention it once in the “Current Study” section. The rest of the Introduction should be a rationale for and review of previous research relevant to the current study, not a description of or transition to it.

2. My comment also still stands for the overly long Results section due in part to repeating numbers in tables (e.g., r and omega coefficients, p-values).

Reviewer #2: The authors have carefully addressed all of my comments, and I have no further concerns about the paper.

However, I have one minor recommendation before publication concerning the following sentence that the authors added on page 9: “At the state level, Edershile et al. (2019) first tested the structure of grandiose and vulnerable narcissism as a multilevel two-factor model, and consequently modeled grandiose narcissism as a model with one factor at both the within- and between-person level (Edershile & Wright, 2020).”

The word “consequently” implies that the choice of Edershile and Wright (2020) to use a one-factor model at both levels of analysis is a direct consequence of Edershile et al.’s first study (2019), in which they used a two-factor model. However, without providing information on the results (e.g., stating that the two-factor model provided a poor fit, if that was the case), the sentence is not clear.

Since correcting this lack of clarity will be easy, no further reviews are needed from my side. I believe the paper is ready for publication.

7. PLOS authors have the option to publish the peer review history of their article (what does this mean?). If published, this will include your full peer review and any attached files.

Reviewer #1: No

Reviewer #2: **Yes: **Anabel Büchner

---

## [Author Response · Author response to Decision Letter 1]

26 Mar 2023

Reviewer #1

1. My first comment is a clarification of my first comment on the original submission (which was not clear to the authors). Rather than mentioning “the current/present study” (and it is clear that the authors are referring to this study and not some other study) multiple times throughout the Introduction, just mention it once in the “Current Study” section. The rest of the Introduction should be a rationale for and review of previous research relevant to the current study, not a description of or transition to it.

Response: Thank you for clarifying this a bit further. In response to this comment, we slightly revised the structure of the introduction. Specifically, as requested, we used the introduction to review previous relevant research in this area and build and explain the rationale for our study, but not to transition to it. We have now added a new section ‘Current study’ where we integrated all relevant descriptions of the work that we are actually undertaking. We believe that this revised structure indeed helped to improve the overall readability of our introduction.

2. My comment also still stands for the overly long Results section due in part to repeating numbers in tables (e.g., r and omega coefficients, p-values).

Response: We have shortened the Results section by removing numbers that were copied from the tables. We also removed some additional clarifications that are not essential to understanding the findings of our study. Note that we would prefer some clarifications (e.g., regarding bifactor indices) given that not everyone will be familiar with the exact meaning of (some of) these statistics.

Reviewer #2

The authors have carefully addressed all of my comments, and I have no further concerns about the paper.

However, I have one minor recommendation before publication concerning the following sentence that the authors added on page 9: “At the state level, Edershile et al. (2019) first tested the structure of grandiose and vulnerable narcissism as a multilevel two-factor model, and consequently modeled grandiose narcissism as a model with one factor at both the within- and between-person level (Edershile & Wright, 2020).”

The word “consequently” implies that the choice of Edershile and Wright (2020) to use a one-factor model at both levels of analysis is a direct consequence of Edershile et al.’s first study (2019), in which they used a two-factor model. However, without providing information on the results (e.g., stating that the two-factor model provided a poor fit, if that was the case), the sentence is not clear.

Since correcting this lack of clarity will be easy, no further reviews are needed from my side. I believe the paper is ready for publication.

Response: Thank you for pointing this out. We revised this section to remove this ambiguity from our text.

---

## [Decision Letter · Decision Letter 2]

5 Apr 2023

Translating trait to state assessment: The case of grandiose narcissism

PONE-D-22-35155R2

Dear Dr. Heyde,

We’re pleased to inform you that your manuscript has been judged scientifically suitable for publication and will be formally accepted for publication once it meets all outstanding technical requirements.

Kind regards,

Srebrenka Letina, Ph.D.

Academic Editor

PLOS ONE

Additional Editor Comments (optional):

The authors addressed all comments and the papers is advised to be accepted.

Reviewers' comments:

Reviewer's Responses to Questions

**Comments to the Author**

1. If the authors have adequately addressed your comments raised in a previous round of review and you feel that this manuscript is now acceptable for publication, you may indicate that here to bypass the “Comments to the Author” section, enter your conflict of interest statement in the “Confidential to Editor” section, and submit your "Accept" recommendation.

Reviewer #1: All comments have been addressed

Reviewer #2: All comments have been addressed

2. Is the manuscript technically sound, and do the data support the conclusions?

Reviewer #1: Yes

Reviewer #2: Yes

3. Has the statistical analysis been performed appropriately and rigorously? 

Reviewer #1: Yes

Reviewer #2: Yes

4. Have the authors made all data underlying the findings in their manuscript fully available?

Reviewer #1: Yes

Reviewer #2: Yes

5. Is the manuscript presented in an intelligible fashion and written in standard English?

Reviewer #1: Yes

Reviewer #2: Yes

6. Review Comments to the Author

Reviewer #1: The authors' revisions have addressed all of my concerns (if incompletely) and I have no further comments on this manuscript.

Reviewer #2: As I already stated in my last review, the paper is ready for publication in my opinion.

7. PLOS authors have the option to publish the peer review history of their article (what does this mean?). If published, this will include your full peer review and any attached files.

Reviewer #1: No

Reviewer #2: **Yes: **Anabel Büchner
